# The Therapeutic Potential of Adipose-Derived Mesenchymal Stem Cell Secretome in Osteoarthritis: A Comprehensive Study

**DOI:** 10.3390/ijms252011287

**Published:** 2024-10-20

**Authors:** Elsa González-Cubero, Maria Luisa González-Fernández, Marta Esteban-Blanco, Saúl Pérez-Castrillo, Esther Pérez-Fernández, Nicolás Navasa, Ana M. Aransay, Juan Anguita, Vega Villar-Suárez

**Affiliations:** 1Department of Neurosurgery, Stanford School of Medicine, Stanford University, Palo Alto, CA 94304, USA; elsagonz@stanford.edu; 2Department of Anatomy, Faculty of Veterinary Sciences, Campus de Vegazana, University of Léon-Universidad de León, 24071 León, Spain; marisagf@hotmail.com (M.L.G.-F.); marta270184@gmail.com (M.E.-B.); saul.cyz@gmail.com (S.P.-C.); esther.per.fern@gmail.com (E.P.-F.); 3Department of Molecular Biology, Faculty of Veterinary Sciences, Campus de Vegazana, University of Léon-Universidad de León, 24071 León, Spain; nnavm@unileon.es; 4Center for Cooperative Research in Biosciences (CIC bioGUNE)-Basque Research and Technology Alliance (BRTA), Parque Tecnológico de Bizkaia, Building 801-A, 48160 Derio, Spain; amaransay@cicbiogune.es (A.M.A.); janguita@cicbiogune.es (J.A.); 5CIBERehd, ISCIII, 28029 Madrid, Spain; 6IKERBASQUE, Basque Foundation for Science, 48011 Bilbao, Spain; 7Institute of Biomedicine (IBIOMED), Faculty of Veterinary Sciences, Campus de Vegazana, University of León-Universidad de León, 24071 León, Spain

**Keywords:** mesenchymal stem cells, osteoarthritis, conditioned medium, secretome, inflammatory cytokines

## Abstract

Osteoarthritis (OA) is a degenerative joint disease characterized by cartilage degradation and inflammation. This study investigates the therapeutic potential of secretome derived from adipose tissue mesenchymal stem cells (ASCs) in mitigating inflammation and promoting cartilage repair in an in vitro model of OA. Our in vitro model comprised chondrocytes inflamed with TNF. To assess the therapeutic potential of secretome, inflamed chondrocytes were treated with it and concentrations of pro-inflammatory cytokines, metalloproteinases (MMPs) and extracellular matrix markers were measured. In addition, secretome-treated chondrocytes were subject to a microarray analysis to determine which genes were upregulated and which were downregulated. Treating TNF-inflamed chondrocytes with secretome in vitro inhibits the *NF*-*κB* pathway, thereby mediating anti-inflammatory and anti-catabolic effects. Additional protective effects of secretome on cartilage are revealed in the inhibition of hypertrophy markers such as RUNX2 and COL10A1, increased production of COL2A1 and ACAN and upregulation of *SOX9*. These findings suggest that ASC-derived secretome can effectively reduce inflammation, promote cartilage repair, and maintain chondrocyte phenotype. This study highlights the potential of ASC-derived secretome as a novel, non-cell-based therapeutic approach for OA, offering a promising alternative to current treatments by targeting inflammation and cartilage repair mechanisms.

## 1. Introduction

Osteoarthritis (OA) is one of the most common diseases of the joints and their surrounding tissues and was recorded as the fourth major cause of disability in 2020 [1]. OA attacks cartilage, causing it to deteriorate. It also leads to alterations in the subchondral bone and synovium, as well as producing underlying bone damage and morphological changes such as subchondral sclerosis, subchondral bone cysts, osteophyte formation, and synovitis [2]. The tissue most affected by OA is hyaline cartilage; the disease alters cell–matrix interactions slowly destroying tissue integrity [3]. Although many aspects of OA pathogenesis are still not fully understood, it is thought that the persistent synthesis of several mediators by articular tissues contributes to tissue deterioration. Inflammation is currently thought to play an important role in the early phases of OA development as well as in its progression [3,4]. Experimentally, it has been shown that the release of certain inflammatory substances such as pro-inflammatory cytokines is one of the main mediators of altered metabolism and increased cartilage catabolism in OA [5]. The activity of cytokines on articular chondrocytes is of significant interest in the study of OA as these signaling molecules are proven to cause cartilage breakdown by upregulating inflammatory or catabolic genes and downregulating anti-inflammatory or anabolic genes [6]. In particular, TNF and IL-1, have been shown to alter the expression of various factors in ways that tend to increase the catabolic activity of chondrocytes, causing cartilage matrix degradation, for example, by reducing levels of aggrecan (ACAN) or type II collagen (COL2A1) [6] and by increasing the production of matrix metalloproteinases (MMPs) [7,8], prostaglandin E2 (PGE2) [9], cytokines, chemokines, reactive oxygen species, and nitric oxide (NO) [4,9,10]. Chondrocytes may revert to a developmental molecular program if these pathways are disrupted, resulting in the production of chondrocyte hypertrophy markers such as COL10A1, MMP13, and RUNX2 [11].

Currently, there is a wide range of therapies for inflammatory pathologies such as OA. Unfortunately, many of these treatments are expensive, non-specific, and none of them can offer drug-free pain relief. In the last decade, however, novel therapies based on mesenchymal stem/stromal cells (MSCs) have emerged as promising candidates to treat OA [12,13] and certain other musculoskeletal problems [14].

Initially, it was thought that the therapeutic effects of transplanted MSCs were a result of MSCs migrating to the damaged region and differentiating to replace dead or damaged cells. However, recent research suggests that the benefits of MSCs are achieved due to their secretion of trophic factors [15]. These factors are present in MSC secretome, also known as the conditioned medium (CM), which contains the paracrine soluble substances and extracellular vesicles (EVs) and these proteins are thought to be responsible for angiogenesis regulation, immunological response, tissue protection, and wound healing. As a result, there could be significant therapeutic benefits in substituting MSCs for the trophic factors found in MSC secretome [16,17].

This study aims to assess the protective effects of secretome from adipose tissue-derived MSCs (ASCs) against inflammation in an in vitro pro-inflammatory model. We examine the impact of secretome on cytokines and factors responsible for inflammation and tissue degeneration in OA.

## 2. Results

### 2.1. ASC-Derived Secretome Downregulates the Expression of NOS2, IL6, MMP13, and TNF

Initially, it was necessary to determine whether ASC-derived secretome had any ability to modulate the expression of certain key mediators of inflammation (*NOS2*, *IL6*, *MMP13* and *TNF*). To this end, qPCR was used to observe the expression of these mediators in our four sample types: non-inflamed chondrocytes, ASCs, TNF-inflamed chondrocytes, and CM-treated TNF-inflamed chondrocytes. Figure 1 shows firstly significant increases in *IL6* and *MMP13* expression for TNF-inflamed chondrocytes compared to non-inflamed chondrocytes and secondly that the treatment of chondrocytes with secretome dramatically reduced the expression of these mediators.

Regarding *TNF* and *NOS2*, a similar pattern was observed, i.e., the expression of these mediators increased in response to TNF exposure and decreased in the presence of secretome.

### 2.2. ASC-Derived Secretome Induces Downregulation of Catabolic Markers and Upregulation of Anabolic Markers

Based on the results obtained by qPCR, further investigation was required to identify the specific genes that might be activated by secretome to give the anti-inflammatory response observed. To this end, a DNA microarray was performed to compare changes in gene expression between two sets of experimental samples: (a) TNF-inflamed chondrocytes and non-inflamed chondrocytes (Cond TNF comparison/control condition) and (b) TNF-inflamed chondrocytes and CM-treated and TNF-inflamed chondrocytes (Cond TNF-CM-treated comparison/Cond TNF).

Beginning with the TNF-inflamed/non-inflamed chondrocyte comparison, screening identified a total of 539 DEGs including 412 positively regulated genes and 128 negatively regulated genes. The analysis was repeated to compare gene expression in the Condrocytes TNF-CM-treated comparison/Condrocytes TNF. Here, screening identified a total of 91 DEGs, including 46 positively regulated genes and 45 negatively regulated genes. DEG functional annotation analysis for these two groups of cells is shown on Figure 2A,B and can be summarized as follows: altered genes were mainly involved in (i) molecular functions related to catalytic activity; (ii) the synthesis of cellular components such as organelles, membranes, and families of hydrolytic enzymes, ligases, and proteases; (iii) cellular processes such as cell communication.

Table 1 shows a summary of the top genes that were significantly differently regulated involved in either anabolism or catabolism in our in vitro model of OA. Within the TNF-inflamed chondrocyte comparison with the control condition group, the genes most positively upregulated were those encoding for *CCL5* (chemokine ligand 5), also called *RANTES* (+270.88-fold) and *CCL2* (chemokine ligand 2), known as monocyte chemoattractant protein-1 (*MCP-1*) (+131.97-fold) they act as a chemoattractant for immune cells, particularly monocytes and T cells. They play a role in recruiting inflammatory cells to the joint and promoting chronic inflammation in OA [18]. It is clear that genes involved in catabolism such as *MMPs* [19] [*MMP3* (+161.50-fold) and *MMP1* (+115.76-fold)], *ADAMTS* (+13.39-fold) and pro-inflammatory cytokines such as IL6 (+136.82-fold) are all upregulated in the TNF-inflamed/non-inflamed chondrocyte comparison while, in contrast, they are downregulated in the CM-treated-TNF-inflamed chondrocyte/TNF-inflamed comparison. Furthermore, for genes involved in anabolism such as *ACAN* (+10.61-fold), *COL2A1* (+10.01-fold), *TIMPs* [*TIMP2* (−10.20-fold) and *TIMP4* (−10.07-fold)] and *Versican* (+10.07-fold), the opposite trend is seen: these genes were downregulated in the TNF-inflamed/ non-inflamed chondrocyte comparison and upregulated in the CM-treated-TNF-inflamed /TNF-inflamed chondrocyte comparison. This highlight, once again, the immunomodulatory potential of ASC-derived secretome.

### 2.3. ASC-Derived Secretome Mitigates TNF-Induced MMP and ADAMTS5 Activity

MMPs and ADAMTS5 are known to have a role in cartilage degeneration and the development of OA [20]. Thus, we assessed the levels of these molecules in cell supernatants to observe the therapeutic effects of secretome on inflamed chondrocytes. In this way, ELISA was used to examine the production of MMPs (MMP1, MMP2, and MMP13) and ADAMTS5 in our different experimental conditions (see materials and methods Section 4). Results are shown in Figure 3. Chondrocytes express endogenous MMPs in basal conditions and inflammation with TNF increased the production of ADAMTS5 and all MMPs tested, with the greatest increase being seen for MMP13. Treatment with secretome reduced the production of MMP1, MMP13 and ADAMTS5 to levels beneath those found in TNF-inflamed chondrocytes with the level of reduction being most pronounced in the case of MMP13. MMP2 levels remained very similar under all experimental conditions.

### 2.4. ASC-Derived Secretome Significantly Reduces Levels of Hypertrophy Markers in Inflamed Chondrocytes

Evidence suggests that the activation of hypertrophic differentiation in articular chondrocytes is a key mechanism in the development of osteoarthritis (OA) [21]. To investigate this process, we analyzed two important markers of hypertrophy: *COL10A1* and *RUNX2*. We conducted three tests to assess how their expression in chondrocytes changes with TNF-induced inflammation and how treatment with secretome might mitigate these changes. These tests included qPCR with bone as a positive control (Figure 4A), Western blot with β-actin as an endogenous control (Figure 4B), and immunocytochemical analysis (Figure 4C,D). Consistently across all analyses, we observed upregulation of *COL10A1* and *RUNX2* in TNF-inflamed chondrocytes and downregulation of these markers in conditioned medium (CM)-treated TNF-inflamed chondrocytes.

### 2.5. ASC-Derived Secretome Increases the Expression of SOX9, COL2A1 and ACAN in TNF-Inflamed Chondrocytes

The expression of specific chondrogenic genes, such as *SOX9*, *ACAN* and *COL2A1*, was analyzed under all experimental conditions. Analysis using qPCR (Figure 5A) shows that compared to TNF-inflamed chondrocytes, CM-treated TNF-inflamed chondrocytes had increased levels of *COL2A1* and *ACAN*. TNF-inflamed-chondrocytes showed reduced the expression of *COL2A1* and *ACAN*, compared to non-inflamed chondrocytes while for CM-treated TNF-inflamed chondrocytes, levels of these genes returned to near normal (non-inflamed) levels. The expression of *SOX9* in TNF-inflamed chondrocyte samples did not change significantly compared to non-inflamed samples; however, for CM-treated TNF-inflamed chondrocytes, the levels far exceeded those seen in non-inflamed samples. In the Western blot analysis (Figure 5B), both inflamed chondrocytes and chondrocytes treated with CM exhibited positive bands for *COL2A1* and *SOX9*. These results indicate the presence of these important markers associated with chondrogenesis and cartilage formation. It is worth noting that in the case of inflamed chondrocytes, the presence of positive bands for *COL2A1* and *SOX9* in the Western blot was unexpected. Immunocytochemical analysis (Figure 5C,D) showed significant decreases in *COL2A1*, *ACAN* and *SOX9* expression for TNF-inflamed chondrocytes compared to non-inflamed samples. For CM-treated TNF-inflamed chondrocytes, however, expression of these three genes was increased compared to levels seen for TNF-inflamed chondrocytes, although the observed increase was significant only for *SOX9.*

### 2.6. ASC-Derived Secretome Increases Chondrocyte Proliferation

Figure 6A,B are fluorescence images showing the influence of TNF and CM-treated TNF-inflamed chondrocytes on chondrocyte proliferation. Comparing cell proliferation rates for non-inflamed chondrocytes, TNF-inflamed, and CM-treated TNF-inflamed chondrocytes (Figure 6B), it is clear that proliferation rates were highest for the CM-treated TNF-inflamed chondrocytes. Furthermore, the inflammatory agent (TNF) appeared to slightly reduce chondrocyte viability; however, the presence of CM stabilized cell viability.

### 2.7. ASC-Derived CM Inhibits NF-κB Translocation

NF-κB is a well-known transcription factor involved in TNF-induced effects on a variety of inflammatory and catabolic mediators. TNF triggers NF-κB translocation into the nucleus and DNA binding, which in turn induces gene transcription thus, we studied the possible regulation of NF-κB translocation by CM. Figure 6C shows a comparison of TNF-induced p65-NF-κB DNA binding in TNF-inflamed chondrocytes and non-inflamed chondrocytes and clearly demonstrates that TNF inflammation increases p65-NFκB DNA binding. Considering Figure 6C,D together, it is possible to see in detail how ASC-derived CM reduces the TNF-induced elevation in P65-NFκB expression and how CM does in fact appear to block NF-κB translocation.

## 3. Discussion

This work provides an in-depth study of the therapeutic effects of secretome or CM on TNF-inflamed chondrocytes and supports the thesis that MSC-derived secretome represents an attractive replacement for traditional and regenerative therapies. The secretome encompasses a diverse array of soluble factors, including growth factors, cytokines, chemokines, and extracellular vesicles (EVs), all of which play pivotal roles in promoting tissue repair, reducing inflammation, and modulating immune responses orchestrating the regenerative and immunomodulatory functions of MSCs, thus secretome has been proposed as a novel free-cell treatment for OA, especially for patients who have not responded well to conventional therapies [22]. ASC secretome therapy presents several potential advantages over existing OA treatments. Unlike traditional pharmacological interventions such as non-steroidal anti-inflammatory drugs (NSAIDs) or corticosteroids, which primarily focus on symptom management, ASC secretome targets multiple aspects of OA pathophysiology [12,13]. Of particular interest in this regard is the way in which secretome appears to be able to regulate pro-inflammatory and catabolic factors in chondrocytes, so providing protection against inflammation [23], a result which is supported by our findings. This multi-faceted approach could potentially offer more comprehensive and long-lasting benefits. Moreover, as a cell-free therapy, ASC secretome mitigates concerns associated with cell-based treatments, such as tumor formation risk or cell viability issues [16,17].

Recent studies have shown that the secretion of inflammatory factors, such as pro-inflammatory cytokines, are critical mediators of altered metabolism and increased ECM catabolism [3,24]. Analysis of our in vitro OA model using qPCR showed that, compared to non-inflamed chondrocytes, TNF-inflamed chondrocytes increased expression of *IL6* and *NOS2*. These results are in keeping with other work which has also shown increased expression of *NOS2* after TNF inflammation [25,26] and our microarray results showed upregulation of several genes included this encoding for IL6 (+136.82-fold), a multifunctional cytokine which is thought to contribute to the inflammatory processes in OA by promoting the production of other pro-inflammatory cytokines such as MMPs and stimulating degradation of collagen leading to cartilage erosion and joint damage [27]. TNF increases MMP production; it also increases levels of either ADAMTS4 or 5, and sometimes both [28,29,30] activating proteolytic cartilage degradation. ELISA was used to examine the expression of three particular MMPs: MMP1, 2, and 13 and ADAMTS-5. As expected then, our findings showed TNF-inflamed chondrocytes expressed more MMP13 than non-inflamed chondrocytes. Our results also showed that treatment with CM can modulate the expression of these factors, as has been seen in other works [22,31]. Microarray results also showed the upregulation of metallopeptidase inhibitor (TIMP2 and TIMP4) in samples of chondrocytes inflamed and secretome treated [32]. This is an important result as MMP13 mediates the direct degradation of ECM components and the activation of other MMPs [33,34]. In the case of ADMTS-5, similar behavior was observed, with levels increasing for TNF-inflamed chondrocytes and some reduction with secretome treatment. Levels of MMP2, however, seemed largely unaffected either by TNF-inflammation or the presence of secretome. The difference in behavior between MMP2 tested may be related to the substrate specificity of MMPs, which in turn could explain the many different damage and cell-dependent inflammatory phenotypes [11]. The cytokine TNF has been shown to upregulate MMPs production in chondrocytes through various pathways including via the activation of: mitogen-activated protein kinase (MAPK), NF-κB, or activator protein 1 (AP-1) [11,19]. Considering NF-κB in particular, when this transcription factor activated in response TNF it is translocated into the cell nucleus where it binds to specific DNA sequences triggering gene transcription. In chondrocytes this process is known play a key role in TNF-induced expression of MMPs, ADAMTSs and, indeed, inflammatory cytokines themselves [29]. The key role of NF-κB and how TNF regulates these inflammatory and catabolic mediators makes it of particular interest to the understanding of OA [35] and thus, we chose to examine the influence of secretome on NF-κB activation. Our results show that secretome significantly reduces nuclear translocation in TNF-inflamed chondrocytes. This suggests that observed reductions in the expression of certain catabolic and pro-inflammatory molecules following treatment with secretome, may be due to secretome’s ability to inhibit NF-*κ*B translocation.

Based on the microarray analysis and considering the TNF-inflamed/non-inflamed chondrocyte comparison (Table 1), of particular interest was the upregulation of the gene that encodes for *CCL5* and *CCL2*, respectively, in the TNF-inflamed/non-inflamed chondrocyte comparison, and in the second comparison treated with secretome, it was the most downregulated. This demonstrates that the secretome is able to modulate the expression of these genes, supporting the findings of previous work [22,23,36].

As OA progresses, chondrocytes lose their differentiated phenotype and begin to behave like the terminally differentiated (hypertrophy-like) chondrocytes seen in bone growth plates. Indeed, it has been shown that various hypertrophy markers, including RUNX2 and COL10A1, are seen at elevated levels both in in vitro models of OA and among individuals with OA [21,37]. The (master) transcription factor for the formation of cartilage is *SOX9* [5,38]. This factor upregulates early chondrogenic genes such as *COL2A1*, *COL11A2*, and *ACAN*; it also enhances the differentiation of mesenchymal cells into chondrocytes and negatively regulates late-stage endochondral ossification [38,39]. Conversely, *RUNX2*, a transcription regulator for type X collagen and established marker for chondrocyte hypertrophy is involved in the calcification and degradation of cartilage matrices and directly implicated in the pathogenesis of osteoarthritis [21,37,40]. Our results indicate that secretome significantly inhibited *RUNX2* expression, which in turn promoted *SOX9* upregulation [41]. The inhibition of *RUNX2* may also be responsible for the observed increases in *COL2A1*, and *ACAN* expression seen in our data.

Two key targets of cartilage degeneration during OA are COL2A1 and the proteoglycan ACAN, the former being degraded by the collagenase MMP13, ADAMTS 4 and 5 [24,34,35]. In agreement with the results of other work of Platas et al. [42], the present study shows clear evidence that the treatment of inflamed chondrocytes with secretome upregulates the expression of both cartilage ECM genes: *COL2A1* and *ACAN*. Indeed, the expression of *COL2A1* and *ACAN* in samples treated with secretome was greater than that seen in either TNF-inflamed samples or non-inflamed samples. These findings suggest that secretome could provide protection against OA. In addition, microarray results showed an upregulation in genes such as *versican*, a protein involved in cell adhesion, proliferation, migration, and angiogenesis; it also plays a central role in morphogenesis and tissue maintenance [43].

## 4. Materials and Methods

### 4.1. Biological Material

The primary cells used in the experimental procedures described here were all sourced from Innoprot^®^ (Pamplona, Spain): Human adipose-derived mesenchymal stem cells (HAdMSC, Ref. P10763) and human chondrocytes (HC, Ref. P10970). Cells are cryopreserved at passage one and delivered frozen. ASC and Chondrocyte Cultivation: Cells were resuspended and proliferated (~1 × 10^6^ cells) in T150 flasks with DMEMc (DMEM, Hyclone^®^, Washington, DC, USA) supplemented with 10% (*v*/*v*) fetal bovine serum (FBS, Hyclone^®^) and 1% (*v*/*v*) penicillin/streptomycin (Hyclone^®^) at 37 °C in a humid atmosphere containing 5% CO_2_.

### 4.2. Secretome or Conditioned Medium (CM) Collection

ASCs were expanded over two passages (~1 × 10^6^ cells) and maintained in DMEMc (DMEM supplemented with 10% (*v*/*v*) FBS, (Merck Darmstadt, Germany) and 1% (*v*/*v*) penicillin/streptomycin, (Merck Darmstadt, Germany) at 37 °C in a humid atmosphere containing 5% CO_2_ until approximately 80% confluency. To avoid possible contamination from factors present in the FBS, a media change was performed. The DMEMc was replaced with serum-free DMEM supplemented with 1% penicillin/streptomycin. After 24 h of incubation under these serum-free conditions, the conditioned medium (CM) was collected. Secretome protein concentration was measured using a Micro BCA Protein Assay Kit^®^, (Merck Darmstadt, Germany) using BSA (Merck Darmstadt, Germany) as a standard following the manufacturer’s instructions. Protein concentration in CM was 50 μg/mL.

### 4.3. In Vitro Model of Chondrocyte Inflammation

Chondrocytes (P2) were cultured in a 6-well plate (~1 × 10^5^ cells). When 80% cellular confluence was reached, samples were treated as follows:Group 1 (non-inflamed chondrocytes): chondrocytes cultured in a mix of serum-free DMEM and 1% penicillin/streptomycin.Group 2 (ASC control samples): ASCs cultured in a mix of serum-free DMEM and 1% penicillin/streptomycin.Group 3 (TNF-inflamed chondrocytes): chondrocytes cultured in a mix of serum-free DMEM and 1% penicillin/streptomycin plus TNF (25 ng/mL).Group 4 (CM-treated TNF-inflamed chondrocytes): chondrocytes were treated with TNF (25 ng/mL) and conditioned medium (50 μg/mL). The CM and TNF were added at the same in the cellular cultures.

After 12 h further incubation, cell samples were collected and analyzed using different assays.

### 4.4. Gene Expression Analysis

#### 4.4.1. Quantitative Real-Time PCR (qPCR)

Total RNA extraction from cell cultures was performed using the GeneMATRIX universal RNA purification kit (EURx^®^, Gdańsk, Poland) following the manufacturer’s instructions. RNA concentration was determined using NanoDrop^®^ ND-1000 UV–Vis spectrophotometer (Thermo Scientific^®^, Waltham, MA, USA). A high-capacity cDNA reverse transcription kit (Applied Biosystems^®^, Waltham, MA, USA) was used to synthesize cDNA from 1000 ng of total RNA, following the manufacturer’s instructions. Primers were designed with the OLIGO7^®^ primer v.7 design tool and are listed in Table 2 and *ACTB* was used as a control for the input RNA level. The qPCR reactions were performed using Power SYBR™ Green PCR Master Mix 2× (Applied Biosystems^®^, MA, USA) in a total volume of 20 μL and on a StepOne real-time PCR system (Applied Biosystems^®^, MA, USA). Target gene expression was calculated by the 2^−ΔΔCt^ method and normalized to the control gene, *ACTB*.

#### 4.4.2. Microarray

A microarray was performed to assess the expression levels of whole transcriptome. The samples used were as described in Section 3. RNA integrity, size, and quantification were assessed using RNA Nano Chips with a Bioanalyser (Agilent Technologies^®^, Santa Clara, CA, USA) and a Qubit^®^ 2.0 Fluorometer (Life Technologies^®^, Carlsbad, CA, USA), respectively. Whole-genome expression characterization was performed using a Human HT12 v4 BeadChip (Illumina Inc.^®^, San Diego, CA, USA). Synthesis of cRNA was performed with the TargetAmp^®^ Nano-g^®^ Biotin-aRNA labeling kit, MA, USA for the Illumina^®^, Epicentre system (catalogue number TAN07924) and subsequent amplification, tagging, and hybridization were performed according to the Whole-Genome Gene Expression Direct Hybridization Illumina Inc.^®^ protocol, CA, USA. Raw data were extracted using GenomeStudio analysis software 2.1 (Illumina Inc.^®^). Subsequently, data were processed and analyzed in the R statistical computing environment using the R packages designed by the Bioconductor project [44,45]. Using the lumi package [44,46], raw expression data were background corrected, log2 transformed and quartile normalized.

Pairwise comparisons of gene expression were performed for each of the sample groups. Data were fitted to a linear model and empirical Bayes moderated t-statistics were calculated using the limma package CA, USA [44]. *p*-values were adjusted by determining false discovery rates (FDR) using the Benjamini–Hochberg procedure [47]. A gene was considered differentially expressed if at least one of its associated probes had an FDR-adjusted *p*-value less than 0.05 and an absolute fold change greater than 2. Differentially expressed genes (DEGs) were loaded into the Panther Gene Ontology database accessed on 20 October 2024 and their cellular components, molecular functions and biological processes were analyzed using the database’s online gene function annotation tools [48].

### 4.5. Nuclear Factor Kappa B (NF-κB) Activity Assay

Chondrocytes were seeded in two Nunc Lab-Tek Chamber Slide Systems (Thermo Fisher Scientific^®^, Waltham, MA, USA) (3 × 10^5^ cells per well). TNF (25 ng/mL) and/or CM (50  μg/mL), depending on the sample type (see Section 3), were then added and samples were incubated for 12 h. After this period, cells were fixed using 2% formaldehyde in PBS (Thermo Fisher Scientific^®^, MA, USA) for 15 min at room temperature before being treated overnight at 4 °C with human antip65-NFB pS529-FITC antibody (Miltenyi Biotech^®^, Radolfzell, Germany). Finally, chamber slides were mounted using Vectashield mounting medium containing DAPI (Thermo Fisher Scientific^®^, MA, USA) for inspection under a confocal microscope (Zeiss^®^, Jena, Germany). NIH Image J v1.31 Software^®^ was used to analyze the fluorescence images of cell nuclei obtained to estimate the total number of positive translocated cells in each sample.

### 4.6. Enzyme-Linked Immunosorbent Assay (ELISA)

After 12 h of stimulation with TNF (25 ng/mL), chondrocyte samples were tested for concentrations of MMP1, MMP2 and MMP13, as well as ADAMTS5, using a specific human MMP-1 (Elabscience^®^, Houston, TX, USA), MMP-2 (Elabscience^®^), MMP-13 (Ray Bio^®^, Peachtree Corners, GA, USA) and ADAMST-5 (Elabscience^®^) ELISA kit, following the manufacturer’s protocols. Measurements were carried out at 450 nm using Multiskan GO multi-plate spectrophotometry (Thermo Fisher Scientific^®^, MA, USA) and concentrations were calculated by comparing them to known standards.

### 4.7. Protein Analysis

#### 4.7.1. Total Protein Extraction and Quantification

Chondrocytes plated in 6-well culture dishes were scraped in 200 μL of RIPA lysis buffer [500 mL stock solution: 1.6 mM NaH_2_PO_4_ (Merck^®^, Darmstadt, Germany), 8.4 mM Na_2_HPO_4_ (Merck^®^, Darmstadt, Germany), 0.1% TritonX-100 (VWR^®^, Pen, USA), 0.1 M NaCl (Ambion^®^, Waltham, MA, USA), 0.1% sodium dodecyl sulphate (SDS; Thermo Fisher Scientific^®^) and ddH_2_O] supplemented with sodium deoxycholate (Merck^®^, Darmstadt, Germany), 1 mM sodium fluoride and 1× protease and phosphatase inhibitor cocktails (Roche^®^, Basilea, Switzerland). Protein concentration was quantified using the Bicinchoninic acid (BCA) protein assay (Bio-Rad^®^, Hercules, CA, USA) in a SpectraMax microplate reader (bioNova Scientific^®^, Madrid, Spain).

#### 4.7.2. Western Blotting Analysis

Samples containing 5 μg total protein were combined with 5× loading buffer [250 mM Tris-HCl (Merck^®^) pH 6.8, 500 mM β-mercaptoethanol (Merck^®^), 50% glycerol (Merck^®^), 10% SDS (Merck^®^), and bromophenol blue (Merck^®^)] in H_2_O and denatured by boiling. Proteins were resolved by SDS-PAGE in 8%, 11% or 15% acrylamide gels, using Mini-PROTEAN Electrophoresis System (Bio-Rad^®^). Fractionated proteins were transferred onto nitrocellulose membranes by electroblotting using a Mini Trans-Blot cell (Bio-Rad^®^). Membranes were blocked with 5% non-fat dry milk in 1× Tris Buffer Saline (TBS) [50 mM Tris, 150 mM NaCl (Merck^®^), pH 8.0) containing 0.1% Tween-20 (Merck^®^) (TBST-0.1%)] and incubated with primary antibodies overnight at 4 °C (Table 3). Horseradish peroxidase-conjugated antibodies were used as secondary antibodies to detect immunoreactive protein bands by Western Lightning Enhanced Chemiluminiscence (ECL) Reagent (PerkinElmer^®^, Waltham, MA, USA) with X-ray imaging [(Fujifilm^®^, Singapore) in a Curix 60 Developer (Agfa^®^, Mortsel, Belgium)].

#### 4.7.3. Confocal Microscopy

Cells were sub-cultured on 8-well Nunc Lab-Tek chamber slide system (Thermo Fisher Scientific^®^, MA, USA) (2 × 103 cells/well). Cells were fixed with 2% paraformaldehyde for 15 min prior to overnight incubation with primary mouse anti-COL10A1 (Santa Cruz Technologies^®^, Dallas, TX, USA), anti-COL2A1, anti-ACAN, anti-SOX9 and anti-RUNX2 antibodies (Cell Signaling^®^, Danvers, MA, USA) (1:100) at 4 °C, after which they were treated with secondary biotinylated anti-mouse antibodies (1:100) (Abcam^®^, Cambridge, UK). Cells were then stained with streptavidin-Alexa 488 and streptavidin-Alexa 568 antibodies (1:100) (Invitrogen^®^, Waltham, MA, USA). Finally, chamber slides were mounted using Vectashield mounting medium (Vector Laboratories^®^, Newark, CA, USA) containing DAPI. After staining, cells were imaged with a confocal microscope (Zeiss^®^).

### 4.8. Statistical Analysis

All experiments were completed in triplicate and results are expressed as the mean ± SD of the three experimental outcomes. Statistical analysis was performed using IBM^®^ SPSS^®^ Statistics v29 (IBM Corp., Armonk, NY, USA). Significant differences among groups were determined using ANOVA followed by post hoc analysis for multiple group comparisons or Student’s *t*-test for two-group comparisons. Results with *p* < 0.05 were considered statistically significant.

## 5. Conclusions

Our results suggest that secretome possesses several properties of potential benefit in the treatment of OA including anti-inflammatory properties, stimulating proteoglycan production, and inhibiting chondrocyte catabolism and the production of certain proteolytic enzymes such as metalloproteases and inflammatory mediators such as TNF, IL6 and NOS2. In addition, secretome appears to have a protective effect for cartilage revealed by its ability to inhibit or decrease levels of hypertrophy markers such as *RUNX2* and COL10A1 and MMP13, increase production of ECM component genes *COL2A1* and *ACAN*, and upregulate expression of the chondrogenic marker *SOX9.* In this way, secretome represents a potentially effective treatment for OA.

## Figures and Tables

**Figure 1 ijms-25-11287-f001:**
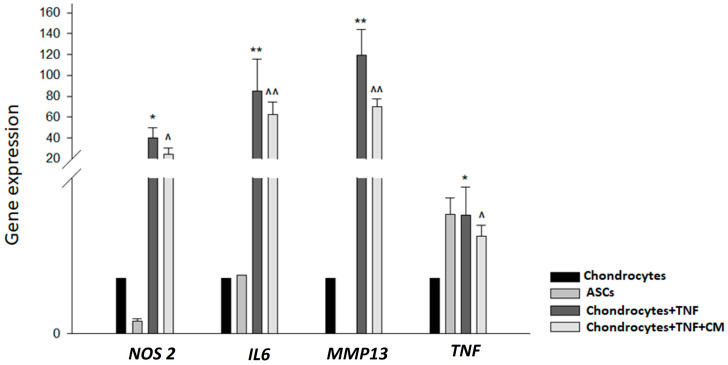
TNF-induced modulation of inflammatory cytokines with and without CM treatment. qPCR was used to measure the relative gene expression of *NOS2*, *IL6*, *MMP13* and *TNF* genes in non-inflamed chondrocytes, ASCs, TNF-inflamed chondrocytes and CM-treated TNF-inflamed chondrocytes after 12 h of incubation. The results are expressed as the mean ± SD of three independent experiments. * (*p* ≤ 0.05), ** (*p* ≤ 0.01) compared to non-stimulated cells ^ (*p* ≤ 0.05), ^^ (*p* ≤ 0.01) compared to TNF-stimulated cells.

**Figure 2 ijms-25-11287-f002:**
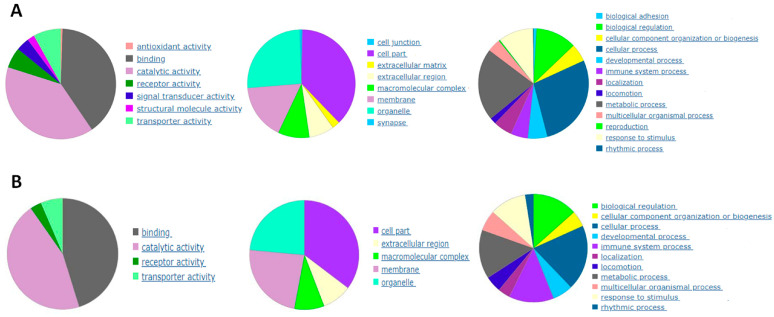
Functional annotation analysis of DEGs that were significantly downregulated or upregulated (**A**) comparing gene expression in non-inflamed chondrocytes and TNF-stimulated chondrocytes and (**B**) comparing gene expression in TNF-inflamed chondrocytes and CM-treated TNF-inflamed chondrocytes.

**Figure 3 ijms-25-11287-f003:**
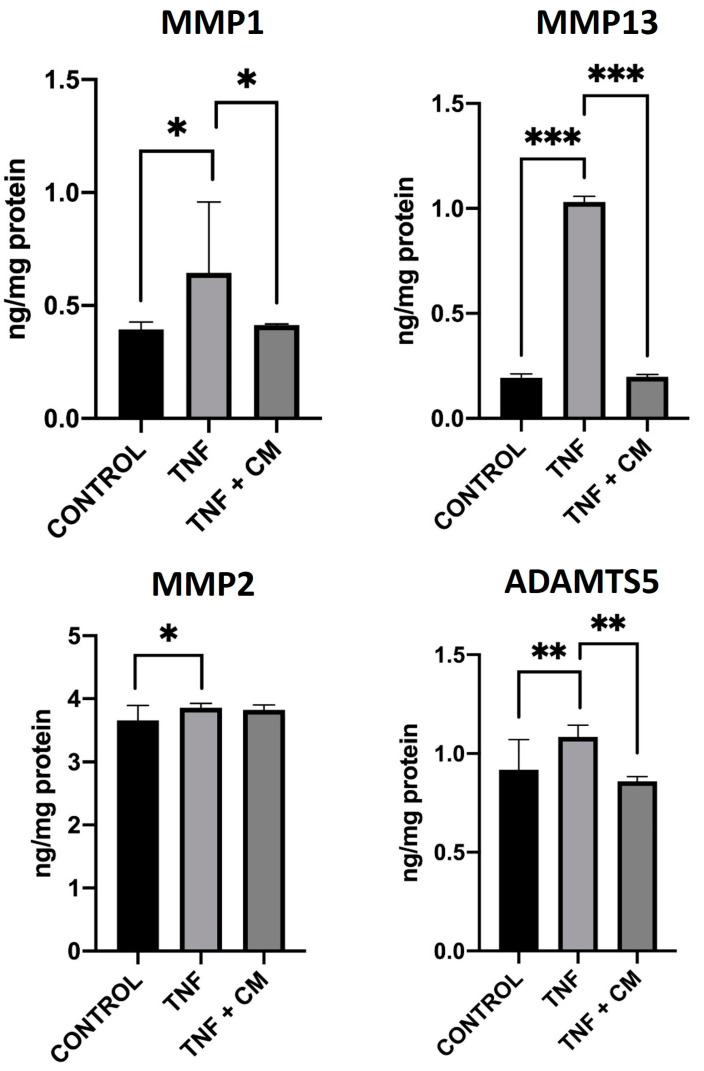
Modulation of MMP1, MMP2, MMP13, and ADAMTS5 proteins in chondrocytes. ELISA was used to measure protein levels comparing non-inflamed, TNF-inflamed, and CM-treated TNF-inflamed chondrocytes after 12 h of incubation. The results are expressed as the mean ± SD of three independent experiments. * *p* ≤ 0.05, ** *p* ≤ 0.01, *** *p* ≤ 0.001.

**Figure 4 ijms-25-11287-f004:**
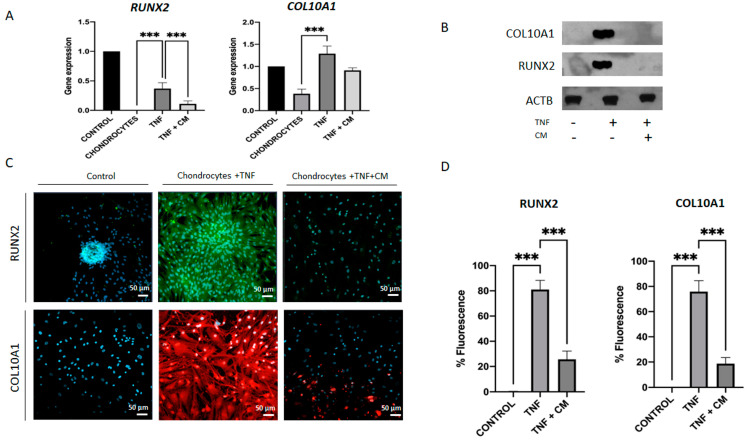
Inflammation-induced upregulation of hypertrophic factors in chondrocytes. *RUNX2* and *COL10A1* gene expression was determined for non-inflamed, TNF-inflamed, and CM-treated TNF-inflamed chondrocytes after 12 h of incubation (**A**) qPCR analysis with bone as control. The results are expressed as the mean ± SD of three independent experiments. *** *p* ≤ 0.001. (**B**) Western blot analysis with ACTB used as endogenous control. Levels of COL10A1 and RUNX2 were detected only in TNF-inflamed chondrocytes not treated with CM. (**C**) Immunocytochemical analysis (representative image): Blue fluorescence indicates cell nuclei (DAPI); green immunofluorescence shows presence of RUNX2 (Alexa-488); and red immunofluorescence indicates the presence of COL10A1 (Alexa-568). Scale bar: 20–50 µm. (**D**) Histogram of RUNX2 and COL10A1 immunofluorescence. The results are expressed as the mean ± SD of three independent experiments. *** *p* ≤ 0.001.

**Figure 5 ijms-25-11287-f005:**
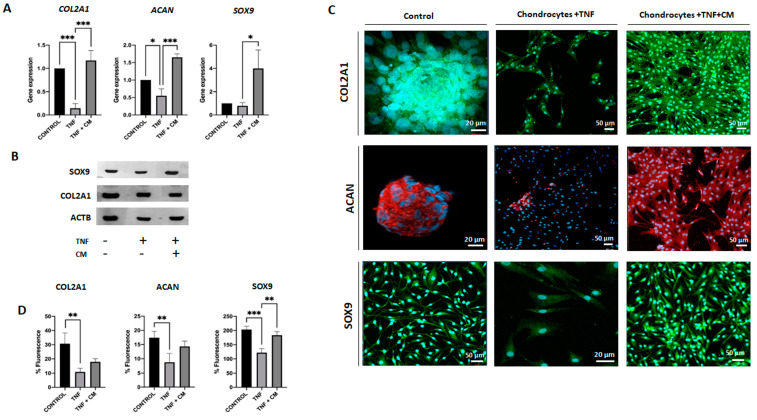
Secretome induced regeneration in inflamed chondrocytes. Gene expression for a range of chondrogenic genes was compared for non-inflamed, TNF-inflamed, and CM-treated TNF-inflamed chondrocytes after 12 h of incubation. (**A**) qPCR analysis of *COL2A1*, *ACAN* and *SOX9*. The results are expressed as the mean ± SD of three independent experiments. * *p* ≤ 0.05, *** *p* ≤ 0.001. (**B**) Western blot analysis of SOX9 and COL2A1; ACTB was used as endogenous control. (**C**) Immunohistochemical analysis of COL2A1, ACAN and SOX9 (representative image). Blue fluorescence indicates cell nuclei (DAPI); green immunofluorescence shows presence of COL2A1 (Alexa-488); red immunofluorescence indicates the presence of ACAN (Alexa-568); and green immunofluorescence shows presence of SOX9 (Alexa-488). (**D**) Histogram showing results for COL2A1, ACAN and SOX9 immunofluorescence. The results are expressed as the mean ± SD of three independent experiments. ** *p* ≤ 0.01, *** *p* ≤ 0.001.

**Figure 6 ijms-25-11287-f006:**
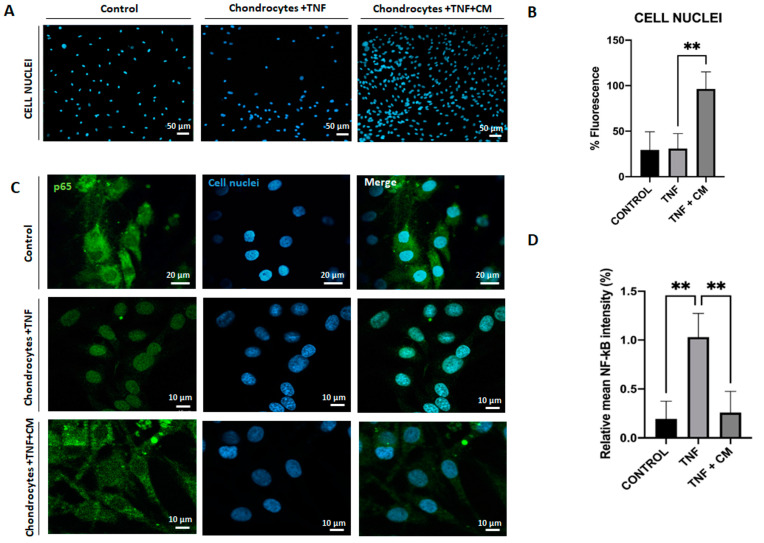
Secretome induced proliferation and blocked NF-κB translocation in inflamed chondrocytes. Non-inflamed, TNF-inflamed, and CM-treated TNF-inflamed chondrocytes were compared after 12 h of incubation to assess cell proliferation and NF-κB activation in the various experimental conditions. (**A**) Cell proliferation (representative images). Blue fluorescence indicates cell nuclei (DAPI). Scale bar: 50 µm. (**B**) Cell proliferation fluorescence histograms. The results are expressed as the mean ± SD of three independent experiments. ** *p* ≤ 0.01. (**C**) NF-κB activation (representative image): Blue fluorescence indicates cell nuclei (DAPI) and green immunofluorescence indicates the presence of NF-κB p65 antibodies so enabling the localization of NF-κB p65. Scale bar: 10 µm. (**D**) Relative intensity of green fluorescence observed cell nuclei. The results are expressed as the mean ± SD of three independent experiments. ** *p* ≤ 0.01 ANOVA was used comparing TNF treated samples with all other groups.

**Table 1 ijms-25-11287-t001:** Genes differentially expressed between TNF-inflamed chondrocytes and control and chondrocytes-TNF inflamed with CM and chondrocytes-TNF inflamed.

	CATABOLISM			ANABOLISM	
Type	Up-Regulated	Down-Regulated	Type	Down-Regulated	Up-Regulated
of Compound	Cond TNF/Ctrl	Cond TNF MC/Cond TNF	of Compound	Cond TNF/Ctrl	Cond TNF MC/Cond TNF
Factor Nuclear Kappa B	NFKBIA (31.84) NFKB2 (18.05) NFKB1 (16.39) NFKBIZ (16.17)	NFKBIA (−12.609) NFKB2 (−10.74) NFKB1 (−10.58) NFKBIZ (−14.00)	Cartilage extracellular matrix proteins	Col2A1 (−10.02) ACAN (−10.63) VCAN (−15.29)	Col2A1 (10.01) ACAN (10.61) VCAN (10.079)
Interleukins	IL1RN (14.38) IL15RA (19.74) IL15 (15.29) IL4I1 (33.98) IL17RB (22.36) IL6 (136.82) LIF (21.39)	IL1RN (−11.77) IL15RA (−12.52) IL15 (−11.52) IL4I1 (−10.48) IL17RB (−14.33) IL6 (−11.70) LIF (−12.39)	Insulin-like growth factors	IGF2R (−10.82) IGFBP6 (−10.95) IGFBP2 (−10.30) IGF2BP2 (−11.52) IGFBP1 (−11.89)	IGF2R (10.05) IGFBP6 (12.20) IGFBP2 (10.11) IGF2BP2 (11.51) IGFBP1 (10.50)
Chemokines	CCL5 (270.88) CCL2 (113.97) CCL20 (62.16)	CCL5 (−17.506) CCL2 (−12.58) CCL20 (−11.84)	Fibroblast growth factors	FGF23 (−10.26) FGFRL1 (−15.04)	FGF23 (10.01) FGFRL1 (1.01)
Metalloproteinases	MMP13 (39.81) MMP1 (115.76) MMP3 (161.50)	MMP13 (−13.78) MMP1 (−1.27) MMP3 (1.73)	Transforming growth factor Beta	TGFB1I1 (−11.13)	TGFB1I1 (1.13)
A desintegrin and metalloproteinase with thrombospondin motifs	ADAMTS5 (13.39) ADAMTS9 (13.28) ADAMTS1 (12.21) ADAMTS13 (10.96) ADAMTS7 (10.04)	ADAMTS5 (−10.40) ADAMTS9 (−11.041) ADAMTS1 (−10.68) ADAMTS13 (−10.32) ADAMTS7 (−10.56)	Platelet-derived growth factors	PDGFA (−10.56) PDGFB (−10.29) PDGFC (−12.19) P DGFD (−10.59)	PDGFA (10.24) PDGFB (10.38) PDGFC (10.54) PDGFD (10.36)
Tumor necrosis factor	TNFSF13B (77.57) C1QTNF1 (71.40) TNFAIP6 (65.74) TNFAIP3 (49.17) TNFAIP2 (21.79)	TNFSF13B (−15.59) C1QTNF1 (−12.40) TNFAIP6 (−14.03) TNFAIP3 (−11.42) TNFAIP2 (−13.16)	Multiple EGF Like Domains 10	MEGF10 (−10.66)	MEGF10 (10.27)
			Vascular Endothelial Growth Factor	VEGFB (−1.12)	VEGFB (10.71)
			Bone morphogenetic protein receptor	BMPR1A (−10.03)	BMPR1A (10.21)
			Interleukin 13	IL13RA1 (−10.92)	IL13RA1 (10.14)
			Metallopeptidase inhibitor	TIMP2 (−10.20) TIMP4 (−10.07)	TIMP2 (10.40) TIMP4 (10.59)
			Mothers against decapentaplegic homolog	SMAD2 (−10.67) SMAD5 (−11.77)	SMAD2 (10.36) SMAD5 (1.045)

**Table 2 ijms-25-11287-t002:** Primer sequences and conditions used for qPCR.

Gen	NCBI RefSeq	Forward/Reverse (5′-3′)	Tª Melting °C	Product Size (bp)
*ACTB*	NM_001101.3	CCCTCCATCGTCCACCGCAAATGCT CTGCTGTCACCTTCACCGTTCCAGT	59.7 58.0	131
*IL6*	NM_000600.3	ATAACCACCCCTGACCCAA CCATGCTACATTTGCCGAA	74.4 72.5	169
*NOS2*	NM_000625.4	AACGTTGCTCCCCATCAAGCCCTT AGCAGCAAGTTCCATCTTTCACCCACT	54.2 54.1	130
*MMP13*	NM_002427.3	CCAGAACTTCCCAACCGTATTGATGC TGCCTGTATCCTCAAAGTGAACAGC	72.3 69.1	145
*TNF*	NM_000594.3	CCTGAAAACAACCCTCAGACGCCACA TCCTCGGCCAGCTCCACGTCCC	77.9 79.3	155
*RUNX2*	NM_001024630.3	AAGCTTGATGACTCTAAACC TCTGTAATCTGACTCTGTCC	55.1 54.0	164
*COL10A1*	NM_000006.12	GCTAGTATCCTTGAACTTGG CCTTTACTCTTTATGGTGTAGG	55.5 56.1	129
*SOX9*	NM_000346.4	AGTTTTGGGGGTTAACTTTG AAGCTTACCAAATGCTTCTC	59.4 57.7	132
*ACAN*	NM_001369268.1	CTGCCCAACTACCCGGCCAT TGCGCCCTGTCAAAGTCGAG	72.1 71.0	200
*COL2A1*	NM_001844.5	CCCATCTGCCCAACTGACC CACCTTTGTCACCACGATCCC	58.5 58.2	166

**Table 3 ijms-25-11287-t003:** Primary antibodies used for protein detection. Target proteins of the antibodies used dilution, and the company is specified.

Target Protein	Dilution	Company
ACTB	1:5000	Merck^®^
SOX9	1:1000	Cell Signaling^®^
RUNX2	1:1000	Cell Signaling^®^
COL2A1	1:1000	Cell Signaling^®^
COL10A1	1:1000	Santa Cruz Biotechnology^®^
ACAN	1:1000	Cell Signaling^®^

## Data Availability

The data used to support the findings of this study are available from the corresponding author upon request. All the microarray data are deposited in the public functional genomics data repository Gene Expression Omnibus (GEO) supporting MIAME-compliant data submissions. To review GEO accession GSE239343: https://www.ncbi.nlm.nih.gov/geo/query/acc.cgi?acc=GSE239343 (accessed on 16 September 2024). Enter token ivevkgmkvfyvler into the box.

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
