# Peer review of "The Therapeutic Potential of Adipose-Derived Mesenchymal Stem Cell Secretome in Osteoarthritis: A Comprehensive Study"

_ijms, 2024, doi:10.3390/ijms252011287_

Round 1
Reviewer 1 Report
Comments and Suggestions for Authors
The manuscript is well-designed and clearly written. However, there are some remarks:
1. Line 65 – „The mesenchymal stem or stromal cells (MSCs)...“ better to write „mesenchymal stem/ stromal cells (MSCs)...“, otherwise it is possible to understand that two different types of cells were used.
2. The nomenclature of genes and proteins should be corrected. All gene names are italicized when referring to the gene, and written in all caps for human genes. Gene and protein names not always are identical. For example genes:
- COL2A1
- ACAN
- SOX9
- COL10A1 (for the Collagen 10, not collagen X)
- TNF
- MMP13
- NOS2 (for the iNOS)
- IL6
- INHBB or INHBA (depending on the used Activin beta subunit), which has to be specified.
While proteins are written in non-italicized form and typically in all caps, though some specific proteins like iNOS and activin β may use mixed case or Greek characters.
- COL2A1
- ACAN
- SOX9
- COL10A1
- TNF
- MMP13
- iNOS
- IL-6
- Activin β or Activin B and so on....
3. The term conditioned medium (CM) is usually used for the cell growth medium with the serum. The clear explanation of used CM is missing in the text. There is one type of CM with serum where cells grow and another CM without serum where also cells grow. Which of them is used for the experiments becomes unclear. Maybe better would be to write, for example, the secretome medium (SM) or serum free conditioned medium (SFCM).
4. Was the secretome medium (here named as CM) added together the TNF at the same time or one after the other? The method part needs more clear explanation.
5. Fig.4 C , Fig. 5C and Fig. 6C – the scale bars and their size should be clearly visible.
6. What about the composition of secretome and factors that could be involved?
Author Response
REVIEWER 1
- Line 65 – „The mesenchymal stem or stromal cells (MSCs)...“ better to write „mesenchymal stem/ stromal cells (MSCs)...“, otherwise it is possible to understand that two different types of cells were used.
Thank you for your correction, it is has been changed and highlighted in the revised manuscript.
- The nomenclature of genes and proteins should be corrected. All gene names are italicized when referring to the gene, and written in all caps for human genes. Gene and protein names not always are identical. For example genes:
You are right, we have changed the genes and protein names in the revised manuscript and the corresponding figures.
- The term conditioned medium (CM) is usually used for the cell growth medium with the serum. The clear explanation of used CM is missing in the text. There is one type of CM with serum where cells grow and another CM without serum where also cells grow. Which of them is used for the experiments becomes unclear. Maybe better would be to write, for example, the secretome medium (SM) or serum free conditioned medium (SFCM).
We agree with this point, we have changed the paragraph to better understanding (Lines 368-374).
“ASCs were expanded over two passages (~1 x 10⁶ cells) and maintained in DMEMc (DMEM supplemented with 10% (v/v) FBS and 1% (v/v) penicillin/streptomycin) at 37°C in a humid atmosphere containing 5% CO₂ until approximately 80% confluency. To avoid possible contamination from factors present in the FBS, a media change was performed. The DMEMc was replaced with serum-free DMEM supplemented with 1% penicillin/streptomycin. After 24 hours of incubation under these serum-free conditions, the conditioned medium (CM) was collected.”
- Was the secretome medium (here named as CM) added together the TNF at the same time or one after the other? The method part needs more clear explanation.
The CM was added at the same time of the TNF. We have clarified in the revised manuscript ( Lines 387 and 388).
- Fig.4 C , Fig. 5C and Fig. 6C – the scale bars and their size should be clearly visible.
You are right, we have improved these images in the revised manuscript.
- What about the composition of secretome and factors that could be involved?
We have included this information in the revised manuscript to clarify this point (Lines 281-285).
“The secretome encompasses a diverse array of soluble factors, including growth factors, cytokines, chemokines, and extracellular vesicles (EVs), all of which play pivotal roles in promoting tissue repair, reducing inflammation, and modulating immune responses orchestrating the regenerative and immunomodulatory functions of MSCs”.
Currently we are developing further studies in our lab in order to characterize the CM.
Reviewer 2 Report
Comments and Suggestions for Authors
This study explores the anti-inflammatory and cartilage-repair effects of adipose-derived mesenchymal stem cell (ASC) secretome in a in vitro model of osteoarthritis (OA). The experimental design is reasonable, and the data analysis is comprehensive. The study is innovative and provides valuable insights into the potential of ASC secretome as a non-cell-based therapy for OA. Below are my review comments:
Strengths:
-
Innovative Research: The study demonstrates the potential of ASC secretome as a novel therapeutic approach for OA, particularly for its anti-inflammatory and cartilage-regenerative effects.
-
Well-designed Experiments: The inclusion of control and treatment groups is appropriate, clearly showing the effects of ASC secretome.
-
Comprehensive Data: A variety of methods (qPCR, ELISA, Western Blot, etc.) were used to validate the findings, and the statistical analysis is well-executed, making the results reliable.
-
High-quality Images: The immunofluorescence staining and other data presentations are clear and well-labeled, aiding in the understanding of the results.
Suggestions for Improvement:
-
Further Analysis of Secretome Components: While the overall effects of the secretome are well-documented, the specific components and mechanisms of action could be further explored or discussed.
-
Statistical Details: Some figures lack details about the number of samples and the number of replicates, which should be added to enhance the robustness of the conclusions.
-
Expand the Discussion: The discussion could benefit from comparing ASC secretome therapy with existing treatment options and outlining its potential advantages and limitations for clinical applications.
Conclusion:
This study provides valuable experimental data supporting the potential of ASC secretome in OA treatment. With minor revisions to address the above points, the paper is suitable for publication.
Author Response
REVIEWER 2
Suggestions for Improvement:
1. Further Analysis of Secretome Components.
Thank you for this consideration, we have included this information in the revised manuscript to clarify this point (Lines 281-285).
“The secretome encompasses a diverse array of soluble factors, including growth factors, cytokines, chemokines, and extracellular vesicles (EVs), all of which play pivotal roles in promoting tissue repair, reducing inflammation, and modulating immune responses orchestrating the regenerative and immunomodulatory functions of MSCs”.
Currently we are developing further studies in our lab in order to characterize the CM.
2. Statistical Details: Some figures lack details about the number of samples and the number of replicates, which should be added to enhance the robustness of the conclusions.
We have reviewed the entire manuscript and in all figure legends where statistics are performed, the following text appears: The results are expressed as the mean ± SD of three independent experiments.
If you consider any additional information important to add, we will change and incorporate it.
3. Expand the Discussion: The discussion could benefit from comparing ASC secretome therapy with existing treatment options and outlining its potential advantages and limitations for clinical applications.
The discussion has been changed and a paragraph comparing traditional therapies for OA with secretome, has been included in lines 281-297.